# Influence of Operating and Electrochemical Parameters on PEMFC Performance: A Simulation Study

**DOI:** 10.3390/membranes13030259

**Published:** 2023-02-22

**Authors:** Imtiaz Ali Soomro, Fida Hussain Memon, Waqas Mughal, Muhammad Ali Khan, Wajid Ali, Yong Liu, Kyung Hyun Choi, Khalid Hussain Thebo

**Affiliations:** 1College of Materials Science and Engineering, Beijing University of Chemical Technology (BUCT), Beijing 100029, China; 2Department of Mechatronics Engineering, Jeju National University, Jeju 63243, Republic of Korea; 3Department of Electrical Engineering, Sukkur IBA University, Sukkur 65200, Pakistan; 4Department of Mechanical Engineering, Quaid-E-Awam University of Engineeirng, Science, and Technology, Nawabshah 67480, Pakistan; 5Institute of Chemical Sciences, Bahauddin Zakariya University Multan, Multan 60800, Pakistan; 6Institute of Metal Research, Chinese Acadmey of Sciences (CAS), Wehua Road, Shenyang 110016, China

**Keywords:** temperature, relative humidity, exchange coefficient, current density, porosity

## Abstract

Proton exchange membrane fuel cell, or polymer electrolyte fuel cell, (PEMFC) has received a significant amount of attention for green energy applications due to its low carbon emission and less other toxic pollution capacity. Herein, we develop a three-dimensional (3D) computational fluid dynamic model. The values of temperature, pressure, relative humidity, exchange coefficient, reference current density (RCD), and porosity values of the gas diffusion layer (GDL) were taken from the published literature. The results demonstrate that the performance of the cell is improved by modifying temperature and operating pressure. Current density is shown to degrade with the rising temperature as explored in this study. The findings show that at 353 K, the current density decreases by 28% compared to that at 323 K. In contrast, studies have shown that totally humidified gas passing through the gas channel results in a 10% higher current density yield, and that an evaluation of a 19% higher RCD value results in a similar current density yield.

## 1. Introduction

Fuel cell technology is considered one of the mature technologies for clean energy applications, and it is the best alternative to replace traditional power systems based on fossils fuels [1,2,3,4,5,6,7,8]. Proton exchange membrane fuel cells (PEMFCs) are advantageous due to their low operating temperature, cleanliness, and ability to keep the environment safe while providing great energy efficiency and density. The performance of PEMFCs might be greatly enhanced by optimizing the structure of each functional layer and operational parameters [9]. A PEMFC consists of four types of essential components, i.e., polymer electrolyte or proton exchange membrane (PEM), catalyst layers (CL), GDL, and channel plates (endplates) at external sides [9,10,11]. The hydrogen gas (H_2_) is oxidized at anode to proton and electron, while at cathode, the oxygen is reduced. The electron moves through the GDLs to the current collector through the external circuit. While wet reactant gases transfer through GDL to the CL, where a chemical reaction takes place. Electrodes are porous, where gases are distributed on the CL [11,12,13,14,15].

The PEMFC usually involves a complicated electrochemical reaction method [4,5,16,17,18,19], which requires expensive equipment and technology. The formation of a suitable mathematical model reached a deep understanding of internal processes, including mass tranport, heat transport, and other chemical reactions that occur in a PEMFC, which can not be directly observed during the reaction [20]. Therefore, many cell configurations and operating conditions can be simulated using these models [21]. Recently, these models, which are based on fuel cells, have received a significant amount attention with the ultimate aim of better understanding the underlying phenomenon of operating fuel cells [22]. Simulation studies allowed the scientific community to gain deep insight and understanding into the highly coupled non-linear chemical and physical aspects of fuel cell technology, as well as prepare design optimization. The author also emphasised the significance of machine learning (ML) and deep learning (DL) for analysing PEMFC’s performance. Among famous modeling tools and methods, the computational fluid dynamics (CFD) is considered as a more powerful tool due to its good capabilities regarding the performance of PEMFC [23]. The performance of PEMFC usually depends on physical factors, including current density, water contents, exchange coefficient, temperature, pressure, and geomtric parameters [24]. Up to date, various 3D PEMFC models were suggested with help of CFD analysis for the pratical implementation of designed optimization and the performance of parameters [25,26,27,28,29,30,31,32,33]. In these models, species transport and water management under typical PEMFC operating conditions are simulated. The results provide detailed information on fluid dynamics and on various other properties, such as chemical, physical, and electrochemical, which exist in fuel cell technology [34,35,36]. Further, the effect of various operating parameters, including pressure, temperature, humidity, etc., on the performance of PEMFC are experimentally performed. The results were compared with their CFD models. The comparison studies showed a good agreement between the model results and the experimental data. This arrangement improves the uniformity of reactant distribution and eliminates the excess water in the prorous electrode, and therfore, improves performance. The CFD model of 50 cm^2^ fuels cells with parallel and serpentine flow fields of different sizes of bipolar plates. It was validated against experimental results, and it was found that the performance of fuel cells is higher at serpentine flow than parallel flow. A study conducted a 3D numerical simulation to verify the proposed optimization models and also compare with the performance of fuel cell using optimized design [37]. Ahmadi et al. [38] demonstated the effect of anode transfer coefficient and species distribution on the performance of cell. It is important to control the gas supply system to improve the response speed and output the power of the cell. 

In this study, a 3D model was developed to investigate the effect of various inlet gas pressures and relative humidity. After a validation with experimental results, the simulation achieved improvement in the performance of cell hydrogen utilization and water management. The parameters selected in this model are temperature, pressure, relative humidity, exchange coefficient, RCD, and GDL porosity. Moreover, this study also investigated the range of electrochemical parameters at both isothermal and non-isothermal conditions on the performance of the cell. Generally, the 7500 RCD value has been considered in other styudies; however, in this study, the RCD value was changed from the multiplication of the RCD value from 0.5 to 3 factor. The CFD commercial software Ansys Fluent Fuel Cell Module was selected to conduct simulations. Because the module has the capability of modeling electrochemistry and current and mass transports of liquid water and heat source in PEMFCs, it is a complete 3D modeling of all the components that represent the structure and geometery of the actual experimental cell. Furthermore, the fluent can also solve basic equations required to model PEMFC. Based on numerical studies, we also optimized the fluid mechanical aspects of cell design to achieve excellent power generation rate and durability.

## 2. Model Development

There are four basic steps in Ansys Fluent for modeling PEMFC (Figure 1). The first step is the geometry of different regions on the fuel cell. The 3D geometry of various parts of the cell is required with defined dimensions. The second step involves setting up the mesh in various regions of the cell, which divides each region into small repeating unit shapes, which fills the entire space in cell. The specific boundries between regions should be defined and also named in the meshing process. After that mesh data tranfered to the fluent, the Ansys Fuel cell module becomes active as an add-on module. The third step involves the input parameters especially for modules that are required. As for the electrodes (anoded and cathode), the reference exchange density, concentration of exponent, reference concentration, exchange coefficient and porosity are needed. In addition, the specific leakage current, open circuit voltage and reference diffusivity of the species must also be entered in an appropriate dailog box. The final step defines the triple-phase boundry layer (catalyst). Besides the porosity, the viscous resistance, solid materials, surface–volume ratio, and contact angle are required. The surface–volume ratio of the catalyst will be used in the Butler–Volmer equation, while the “electrolyte zone” can be chosen as “membrane”. At this stage, a few parameters, such as conduction coefficient, equivalent weight, and protonic conduction exponents, are required. The setting of both the anode and cathode will be the same. Ansys Fluent 15.0 will be used to solve all governing equations.

### 2.1. Geometry, Mesh, and Other Parameters

After understanding the steps of simulation, the geometry of fuel cell membrane, GDL, CL, and flow channels has been drawn by the Ansys Fluent PEMFC Model, as shown in Table 1.

### 2.2. Boundary Conditions

The boundary conditions for design models are:The mass flow inlet as a boundary type is used at inlets of both the anode and cathode sides. The values of temperature, mass flow rate, and mass fraction of hydrogen, oxygen, nitrogen gases, and water are prescribed;At the outlets on both sides of the anode and cathode, the boundary condition is the pressure outlet used. While the value of pressure is prescribed and the options of backflow conditions are defined;The stationary wall type shall be set as a wall region between outlets and inlets;There is zero flux boundary condition for a membrane phase potential on all outside boundaries because no photonic current leaves the fuel cell through an external boundry;On external contact boundaries, the values of solid potential are fixed as potential static boundary conditions. In this model, the solid phase potential is fixed as the potential on the anode side and on the cathode side are set to zero. The solid-phase potential is set to the cell voltage.

### 2.3. Model Validation

The model validation, including numerical and physical, is highly desirable for practical applications [32,33]. To investigate the performance of PEMFC, the voltage current density or polarization curve are the most important outputs of numerical simulations. The simulation results for base-case operating conditions were verified with the reported experimental data. Figure 2 shows the computed polarization curves, which are well matched with experimental curves in low-load regions. However, the current density model is the high-mass tranport limited region, which is higher than the obtained experimental results. Therefore, this type of observation is common in those models where there is an effect of reduced oxygen transport because of water flooding in cathode at high current density. 

### 2.4. Model Assumptions

The following assumptions are adopted in this model, such as that PEMFC operates at a steady state. The gases species are ideal gasses, while flow is laminar, and CL, GDL, and MEA are isotropic porous layers. Water produced in the cathode CL is in the vapour phase. 

## 3. Governing Equation

Using electrochemistry models, researchers may estimate local current densities and voltage distributions in PEM fuel cell catalyst layers [14]. The anode and cathode reaction rates are computed using Butler–Volume equations in electrochemistry models. One potential equation deals with the electron transport through a solid conducting substance, and the other potential equation deals with the protonic transfer of H^+^. From Ansys Fluent 2012 [15], all of these equations have been used in this model. The electron flow through solid materials, such as catalysts and diffusion layers, is characterized by the following equation
(1)$$\nabla \;\cdot \;(\sigma_{sol}\nabla {Ø}_{sol})+R_{sol}=0$$

Equation for the flow of protons through the membrane is also defined as
(2)$$\nabla \;\cdot \;(\sigma_{mem}\nabla {Ø}_{mem})+R_{mem}=0$$
where

σ = electrical conductivity (I/ohm.m)

φ = electrical potential (volts)

*R* = volumetric transfer current (A/m^3^)

The main Butler–Volume functions are used to define the current density source terms *R_mem_* and *R_sol_*
(3)$$R_{\mathrm{an}}=\left(\xi_{an}j_{an}^{ref}\right){\left(\frac{\left[A\right]}{{\left[A\right]}_{ref}}\right)}^{\gamma_{an}}\left(-e^{+\alpha_{anF\eta_{an}/RT}}+e^{\alpha_{catF\eta_{an}/RT}}\right)$$
(4)$$R_{\mathrm{cat}}=\left(\xi_{cat}j_{cat}^{ref}\right){\left(\frac{\left[C\right]}{{\left[C\right]}_{ref}}\right)}^{\gamma_{cat}}\left(-e^{+\alpha_{anFcat/RT}}+e^{\alpha_{catFcat/RT}}\right)$$
where

*j^ref^* = refers to the exchange current per active surface area (A/m^3^)

ζ = specific surface active area (1/m^2^)

Y = concentration-dependent exponent

α = exchange coefficient

*F* = Faradays Constant

*R* = General gas constant (8.3142 J/mole.k)

Species volumetric source terms in the catalyst layer are called triple phase boundaries (TPB), due to electrochemical reactions for PEM fuel cell being
(5)$$S_{H_{2}}=-\frac{M_{wH_{2}}}{2F}R_{an}<0$$
(6)$$S_{O_{2}}=-\frac{M_{wO_{2}}}{4F}R_{cat}<0$$
(7)$$\;S_{H_{2}O}=\frac{M_{wH_{2}O}}{2F}R_{cat}>0$$

During the electrochemical reaction, chemical energy is converted into electrical work due to the irreversibility of the process; the net available energy is given by
(8)$$\;\;S_{h}=h_{react}-R_{an,cat}\eta_{an,cat}+I^{2}R_{ohm}+h_{L}$$

Where

*h_react_* = net enthalpy change during chemical reaction

*R_an,cat_ ɳ_an,cat_* = product of transfer current and over potential in the anode or cathode.

*H_L_* = enthalpy change due to the condensation vaporization of water

*I* = total current

The liquid water formation and transport is governed by the following equation for the volume fraction of liquid water, s, or the water saturation
(9)$$\frac{\partial \left(\varepsilon \rho_{t}s\right)}{\partial_{t}}+\nabla \;\cdot \;\left(\rho_{t}\vec{V_{t}}s\right)=r_{w}$$
where
(10)$$r_{w}=C_{r}max\left(\left[\left(1-s\right)\frac{P_{wv}-P_{sat}}{RT}M_{w,H_{2}O}\right],\left[-s\rho_{t}\right]\right)$$

The back diffusion flux of water is modeled as
(11)$$\int_{w}^{diff}=-\frac{\rho m}{M_{m}}M_{H_{2}O}D_{i}\nabla \lambda$$
where *ρ_m_* and *M_m_* are equivalent weights of dry membrane.

The membrane water diffusibility is modeled as
(12)$$D_{i}=f\left(\lambda \right)e^{2416\left(\frac{1}{303}-\frac{1}{T}\right)}$$

The water content (λ) of the membrane is modeled as
(13)$$\lambda =0.043+17.18\alpha -39.85\alpha^{2}+36\alpha^{3}\;(\alpha <1)$$
(14)λ=14+1.4(α−1) (α>1)
where α is water conductivity and ia defined as
(15)$$\alpha =\frac{P_{wv}}{P_{sat}}+2s$$

All governing equations are solved through Ansys Fluent 15.0 with its PEMC add-on module.

## 4. Results and Discussion

The base case was used to perform a series of simulations from low to high operating current density. The polarization curves show agreement between the predicted results and experimented data published in the literature [39]. Electrochemical parameters in the Butler–Volmer equation, such as RCD, exchange coefficient and concentration exponents, are electrode- and catalyst-dependent, while the electrochemistry of the cell is controlled by the cathode oxygen reduction reaction (ORR). The values of the exchange coefficient and current density are considered as model parameters for the calibration of numerical results along with the experimental measurements. Simulations were performed for the values of the parameters, as shown in Table 2. An Ansys Fuel Cell module was applied to run the simulation and the results were listed to draw the V-J curves.

### 4.1. Effect of Temperature

The operating temperature is the initial temperature at which PEMFC operates. The temperature seriously affects all transport processes in the cell. The composition of incoming gas stream also depends on the temperature. The molar oxygen fraction at the CL decreases with the increasing cell-operating temperature because of the reduction in the molar oxygen fraction in the gas stream. Simulations of temperature were run at 323 k to 353 k. The polarization curves of the results are shown in Figure 3. These curves show that the performance of the cell increases with the incease of temperature. The exchange current density (ECD) also increases with the increase in the operating temperature, which reduces action loss. This may explain the improvement of the performance of PEMFC [39].

### 4.2. Effect of Operating Pressure

Operating pressure refers to the initial pressure at which PEMFC operates. In general, a PEMFC is operates at ambient (atmospheric) pressure, although the cell may be pressurized, and thus, it can operate at any pressure [40]. The polarization curves are obtained by operating PEMFC from 1 to 4 atm, as shown in Figure 4. It has been found that an increase in cell-operating pressure, results in higher cell current density. The pressure of the anode and cathode sides had always been kept the same. The performance of the fuel cell improves with the increase in pressure, which is explained by the Nernst equation [41]. Overall, polarization curves shift positively as the pressure increases. Another reason for the improved performance is the partial increase in reactant gases with the increasing operating pressure [39].

### 4.3. Effect of Anode- and Cathode-Relative Humidification

When PEMFC is sufficiently humidified, the performance of cell can be improved with an incease in temperature. The performance of PEMFC is significantly affected, when the operating temperature is greater than the humidification temperature, while the cathode humidification temperature does not have any effect on the performance of the cell at higher current densities [39]. Simulation studies were performed for relative humidity values 25%, 50%, 75%, and 100% at the anode and cathode sides. According to the simulated result, as shown in Figure 5, when the anode-relative humidity increases, the overall water uptake in the system increases. This leads to an improvement of the cell performance. Water transportation takes place in the membrane through an electro-osmotic drag, and the back diffusion is affected by the membrane, current density, water content, operating temperature and humidity of the reactant gases. Therefore, to obtain the ideal perfomance, there must be water balance between the anode and cathode. When the fuel at anode inlet is fully humidified, the humidity of the membrane can be mainatined [42].

The flow of protons and the amount of water produced at the cathode increases linearly as the current density increases. The reaction at the cathode side accumulates water, resulting in the nanopore of the membranes being blocked and also to an inhibited tranport of species when the gases are fully humidified at the inlet to 100%. Further, the current density is significantly improved by decreasing the relative humidity of the cathode at a low cell volatge. It can be concluded that the cathode-relative humidity does not have a more dominant effect on the performance of the cell than anode relative humidity [43].

### 4.4. Effect of Exchange Coefficient

Simulation studies were carried out for four different values of exchange coefficient. Theoretically, the exchange coefficient can not be larger than 2. Therefore, the values of coefficient were selected 0.5, 1.0, 1.5, and 2 for both the anode and cathode. The polarization curves are shown in Figure 6. Studies showed that as the exchange efficient decreases, the V-J curve is decreased. The slope of curves for exchange coefficient of 0.5 and 1 are different, while others show similar values as current density increases. Since the exchange coefficient appears to be a positive multiplying factor in the exponent of the dominant term in the Butler–Volmer equation, this change in the slope with the decreasing exchange coefficient is to be expected. In addition, when the exchange coefficient is larger, the curve moves closer to a straight line. The exchange coefficients are considered the most important kinetic parameters for electrode reactions. They are related to the type and specification of the surface of the electrode and properties of the catalyst [23,44]. 

### 4.5. Effect of RCD

The default values of RCD in the Ansys Module are 7500 A/m^3^ on the anode side and 20 A/m^3^ on the cathode side. These values were multiplied by 0.5, 1, 1.5 and 2 to run the simulation. Results are given in the V-J curves, as shown in Figure 7. From the Butler–Volmer equation, it is expected that the increasing ECD raises the V-J curve (lifts the current density for a given voltage). As for the higher range of the current density, all V-J curves of all RCDs are approximately parallel. RCD is a measure of readiness of the electrode to proceed with the electromagnetic reactions. If the reference current is high then the surface of the electrode is more active. On the anode hand, the RCD is greater than the cathode side. The higher the RCD value, the more the energy barrier decreases, so that the changes that must be overcomed are moving from the electrolyte to the surface of the catalyst and vice versa [21]. In simple terms, the higher the value of RCD, the more current is produced. 

### 4.6. Effect of GDL Porosity

The values of GDL porosity on both the anode and cathode sides were selected as 0.2, 0.4, 0.6, and 0.8, as shown in Table 2. The simulation results are shown in Figure 8, and the results are exhibited so that the current density is increasing with higher values of porosity. It can be predicted that the performance of PEMFC can be ideal at high values of porosity. The porous region of GDL provided the space for the reactant to move towards the catalyst region. The increase in porosity means that the onset of mass transport limitations occurs in high current densities, such as leading to high currents [40,45]. The PEMFC with the lowest porosity value of 0.2 had a high mass transfer resistance of reaction because of the small pores in GDL.

### 4.7. Effects of Electrochemical Parameters at Non-Isothermal Conditions

The simulations are conducted at non-isothermal conditions for different values of RCD at a temperature from 323 k to 353 k. The polarization curves are shown in Figure 9, and it is shown that the maximum current density occurs at the 353 k of each simulation result of the RCD, and its peak value is 3. The obtained high value of the ECD indicated that the electrode surface is highy active for electrochemical reactions. The total current density at both the anode and cathode electrodes must equal the conservation of charge. The values of ECD must be higher by several orders of magnitude than at the cathode [46]. The effect between RCD and temperature has been shown in Figure 9d. The curve shows that both are increasing. 

### 4.8. Effect of GDL Porosity on Exchange Coefficient

Simulations were conducted by varying temperature values from 323 k to 353 k for each of the GDL porosity values 0.2, 0.4, 0.6, and 0.8, and the curves are shown in Figure 10. In Figure 10a–c it is shown that the maximum current density is occurring at the maximum value of GDL porosity and the highest value of current density occurs at the porosity values of 0.8 and 1.5 exchange coefficients. Figure 10d shows the effect between the exchange coefficient and GDL porosity. Saderberg [47] investigated the influence of the properties of the porous electrode on the value of the charge transfer coefficient. The results show that the value of the charge transfer coefficient increases with an increase in electrode porosity. This confirms our simulation results.

**Table 2 membranes-13-00259-t002:** Operating and electrochemical parameters of PEMFC.

Paramters	Anode Value	Cathode Value	Ref.
RCD (A/m^3^)	3750, 7500, 15,000, 22,500	10, 20, 30, 40	Assumed
Exchange coefficient	0.5, 1, 1.5, 2	0.5, 1, 1.5, 2	Assumed
Pressure (atm)	1, 2, 3, 4	1, 2, 3, 4	[39]
Temperature (K)	323, 333, 343, 353	323, 333, 343, 353	[39]
Relative humidity	25%, 50%, 75%, 100%	25%, 50%, 75%, 100%	[42]
GDL porosity	0.2, 0.4, 0.6, 0.8	0.2, 0.4, 0.6, 0.8	[45]
Concentration exponent	0.5	1	[48]
Reference concentration (K mol/m^3^)	1	1	[48]
CL porosity	0.5	0.5	[48]
Mass flow rate H_2_ (Kg/s)	6.0 × 10^−7^	5.0 × 10^−6^	[48]
Open circuit voltage (V)	0.95	0.95	[48]
GDL porosity	0.2, 0.4, 0.6, 0.8	0.2, 0.4, 0.6, 0.8	[49]

## 5. Conclusions

The 3D CFD model has been developed by using commercial software Ansys Fluent to investigate the performance of PEMFC at varying values of input operating and electrochemical parameters, such as temperature, pressure, relative humidity, exchange coefficient, RCD, and GDL porosity. The simulations were also conducted at varying temperature value from 323 to 353 k and pressure values from 1 to 4 atm. The voltage of the anode side was kept constant at 0 V, whereas it was varied from 0.4 to 0.9 V on the cathode side. The temperature and pressure were kept at constant values, i.e., 353 k and 3 atm, respectively, to study the effects of variations in the rest of the parameters. The variations of input parameters have shown an influence on the overall performance of PEMFC. In this work, it has been invistigated that 28% of the current density has been lost on the 353 K of temperature. Similarly, higher current density has been evaluated on the 22,500 RCD value, which is three times higher than the default RCD value at the anode side. According to the findings of the research, the greatest value of the current density appears at the porosity values of 0.8 and 1.5 of the exchange coefficient, and maximum current density occurs at the maximum value of the GDL porosity. The high value of ECD that was found suggests that the electrode surface was very active for the electrochemical process investigated. It is becuase of non-isothermal condition and RCD value. It has been found that by increasing the values of parameters, the current density is also increasing. Furthe, increase in operating temperature and pressure results in the enhancement of cell performance. High temperature improves the water management and boosts electrochemical reaction rate in PEMFC, whereas high pressure increases the concentration rate of reactant gases. Anode humidification has a more significant effect on performance than cathode humidification. By changing the values of exchange coefficient, the RCD, and GDL porosity, the Ansys Fuel Cell Module has shown the effect on the performance of the fuel cell. The results of the simulation conducted at the non-isothermal conditions of the electrochemical parameters provide insight on the influence of temperature on exchange coefficient and RCD. Finally, the impact of GDL porosity on the exchange coefficient has also been investigated through simulation results. This study makes several noteworthy contributions towards important parameters involved to enhance the performance of PEMFC.

## Figures and Tables

**Figure 1 membranes-13-00259-f001:**
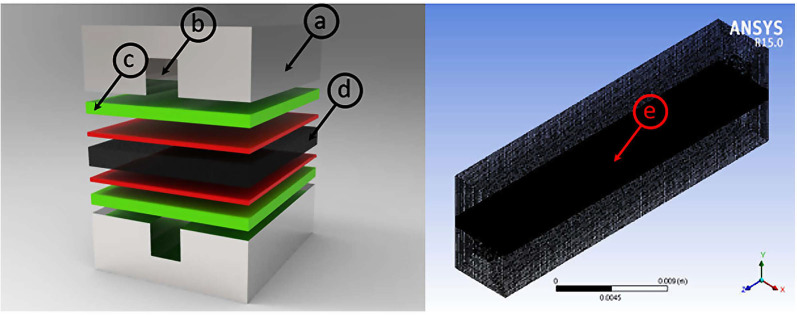
3D model of different parts of PEMFC: (**a**) Current flow channel; (**b**) Cathode and anode gas channel; (**c**) GDL; (**d**) Membrane; (**e**) 3D mesh model for numerical simulation.

**Figure 2 membranes-13-00259-f002:**
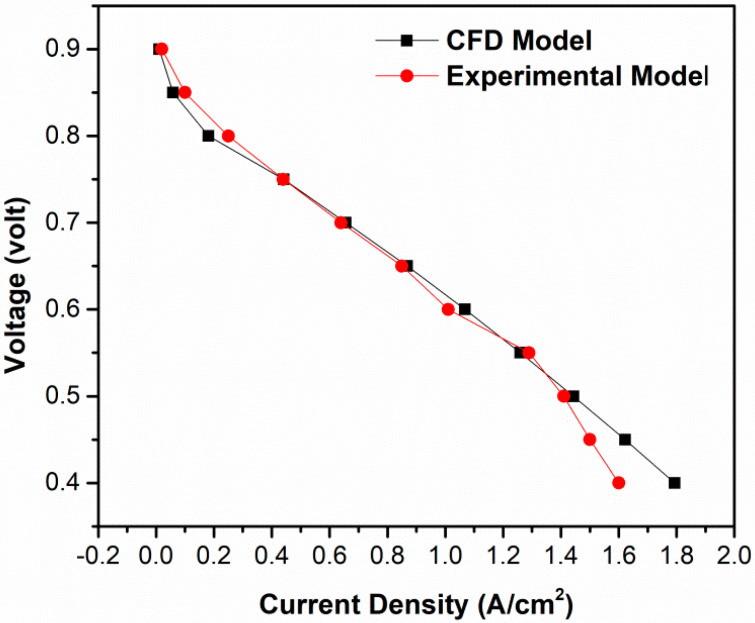
Polarization curves for CFD and experimental model.

**Figure 3 membranes-13-00259-f003:**
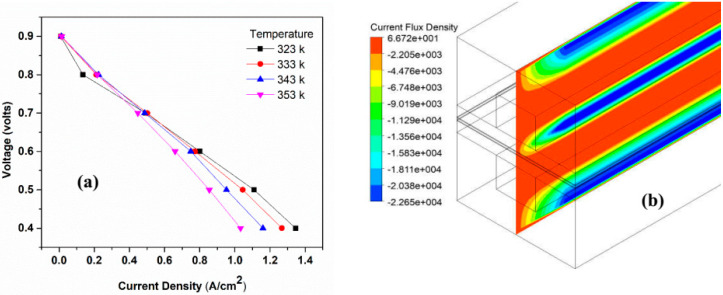
(**a**,**b**) Effect of temperature on performance of PEMFC: (**a**) Polarization curve of different temperature; and (**b**) Contours of current density magnitude at 0.6 V for 353 k.

**Figure 4 membranes-13-00259-f004:**
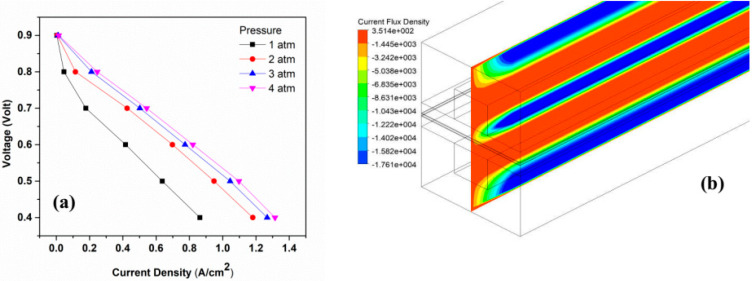
Effects of pressure on performance of PEMFC: (**a**) Polarization curves for pressure; and (**b**) Bontours of the current flux density at 0.6 V.

**Figure 5 membranes-13-00259-f005:**
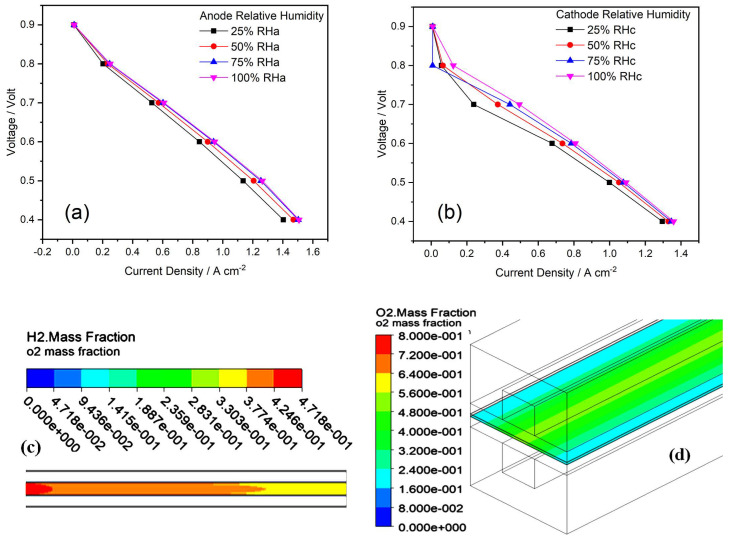
(**a**–**d**) Effect of relative humidity on performance of PEMFC: (**a**,**b**) Polarization curves of relative humidity at anode (**a**) and cathode (**b**), respectively. (**c**,**d**) Contours of mass fraction of H_2_ (**c**) and O_2_ (**d**), respectively.

**Figure 6 membranes-13-00259-f006:**
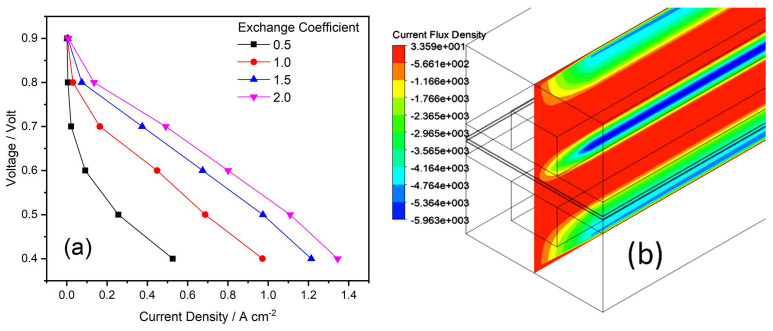
(**a**,**b**) Effects of exchange coefficient on PEMFC performance: (**a**) Polarization curves of exchange coefficient; and (**b**) Contours of the current flux density at 0.6 V for exchange coefficient value of 0.5.

**Figure 7 membranes-13-00259-f007:**
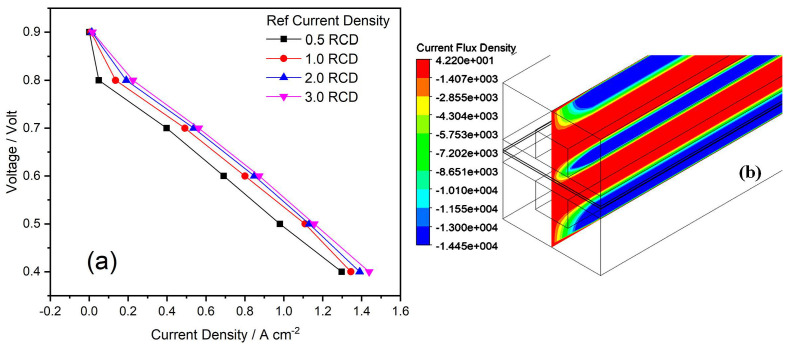
(**a**,**b**) Effects of RCD on PEMFC performance: (**a**) Polarization curves of RCD; and (**b**) Contours of RCD at 0.6 V.

**Figure 8 membranes-13-00259-f008:**
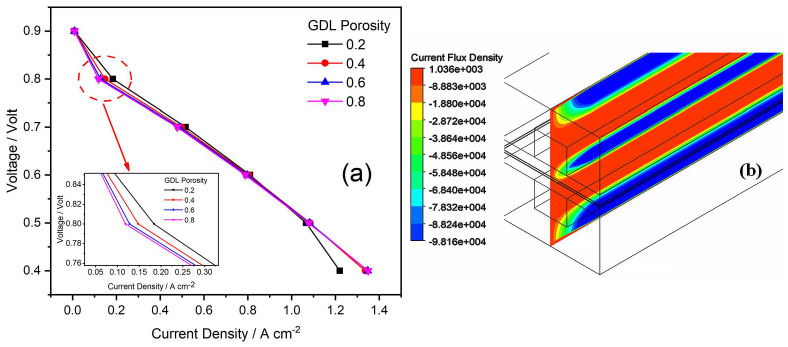
(**a**,**b**) Effects of GDL porosity on PEMFC performance: (**a**) Polarization curves of GDL porosity; and (**b**) Contours of the current flux density at 0.6 V for GDL porosity.

**Figure 9 membranes-13-00259-f009:**
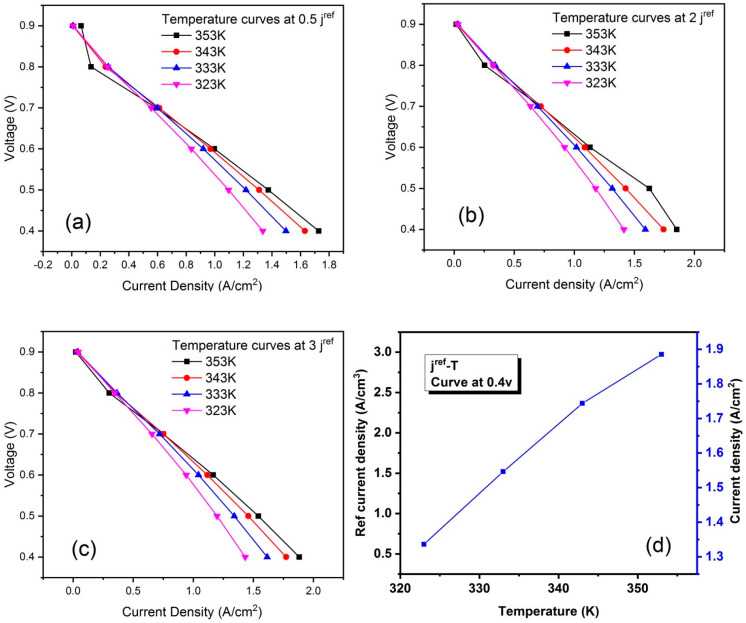
Effects of temperature on RCD at (**a**) 0.5; (**b**) 2; (**c**) 3; and (**d**) Polarization curve between RCD and temperature at 0.4 v.

**Figure 10 membranes-13-00259-f010:**
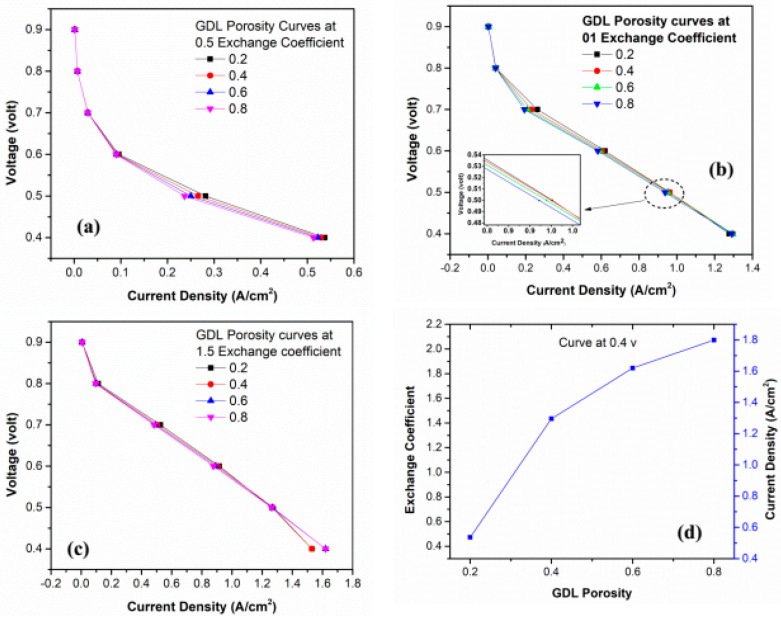
GDL porosity curves at exchange coefficient’s values: (**a**) 0.5; (**b**) 01; (**c**) 1.5; and (**d**) Polarization curve between exchange coefficient and GDL porosity at 0.4 v.

**Table 1 membranes-13-00259-t001:** Dimensions of PFMFC geometry.

Part	Length (mm)	Width (mm)	Height (mm)
GDL	25	6	0.3
CL	25	6	0.02
Membrane	25	6	0.15
Channels	25	2	2

## Data Availability

The data presented in this study are available on request from thecorresponding author.

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
