# Peer review of "Influence of Operating and Electrochemical Parameters on PEMFC Performance: A Simulation Study"

_membranes, 2023, doi:10.3390/membranes13030259_

Round 1
Reviewer 1 Report (New Reviewer)
The author developed a three-dimensional computational fluid dynamic model, and investigated the influence of the operation parameter on the PEMFC performance. However, some issues still need to be improved:
Comment 1. Additionally, your literature survey needs improvement; please expand it. A list of possible articles is provided below:
https://doi.org/10.1016/j.ijhydene.2022.02.023
https://doi.org/10.1016/j.energy.2021.122270
Comment 2. The abstract needs to rewrite. It is suggested that the abstract should reflect significance of your work and novelty.
Comment 3. In section 2, the equations related to the simulation model should be added in the manuscript.
Comment 4. In section 2.3, the reference of the reported experimental data hasn’t posted in the text, please supply the related information.
Comment 5. In table 1, are you sure the unit of the dimension parameters is nm? Is the height of the channel only 2 nm??
Author Response
The author developed a three-dimensional computational fluid dynamic model, and investigated the influence of the operation parameter on the PEMFC performance. However, some issues still need to be improved:
Reply: We thank the reviewer very much for comments and suggestions to improve our work.
Comment 1. Additionally, your literature survey needs improvement; please expand it. A list of possible articles is provided below:
https://doi.org/10.1016/j.ijhydene.2022.02.023
https://doi.org/10.1016/j.energy.2021.122270
Reply: Thanks for suggetsions. These papers helped us a lot to improve our MS. We also cited these papers in our MS.
Comment 2. The abstract needs to rewrite. It is suggested that the abstract should reflect significance of your work and novelty.
Reply: We thank the reviewer very much for comments. The abstract is modified.
Comment 3. In section 2, the equations related to the simulation model should be added in the manuscript.
Reply: We thank the reviewer very much for comments. We have revise this section and added mofre equations into revised MS.
Comment 4. In section 2.3, the reference of the reported experimental data hasn’t posted in the text, please supply the related information.
Reply: We thank the reviewer very much for comments. We have modified as per suggestions.
Comment 5. In table 1, are you sure the unit of the dimension parameters is nm? Is the height of the channel only 2 nm??
Reply: We thank the reviewer very much for comments. The text and units are corrected.
Reviewer 2 Report (New Reviewer)
This paper studies the influence of operating and electrochemical parameters on PEMFC performance through simulation experiments, which has potential application value in engineering. In order to meet the requirements of high-quality publication of the "membranes" journal, it is recommended to consider the following suggestions.
1) There is no quantitative data in Abstract Section.
2) Introduction Section needs to be rewritten. In addition to the operating and electrochemical parameters , is other method feasible for improving PEMFC performance? What are the advantages and disadvantages of each? You need to give your own analysis in this section. The following references may have some value and significance, so you can consider quoting them.
[1] Review: Modeling and Simulation of Membrane Electrode Material Structure for Proton Exchange Membrane Fuel Cells[J]. Coatings, 2012, DOI: 10.3390/coatings12081145
[2] A systematic review of machine learning methods applied to fuel cells in performance evaluation, durability prediction, and application monitoring November[J].International Journal of Hydrogen Energy, 2022, DOI: 10.1016/j.ijhydene.2022.10.261
[3] Evolution of atomic-scale dispersion of FeNx in hierarchically porous 3D air electrode to boost the interfacial electrocatalysis of oxygen reduction in PEMFC[J], NANO ENERGY, 2020, DOI: 10.1016/j.nanoen.2020.105734
3) The innovation of this article is not reflected in the first section and needs to be modified.
4) More details need to be demonstrated in Model Development Section.
5) The method proposed in this paper needs to be compared with the previous literature, otherwise it cannot reflect innovation.
6) The Discussion Section needs a separate section.
7) There is no quantitative data in the Conclusion Section.
8) There are few references in the last three years.
9) There is a lack of references in this journal.
Author Response
This paper studies the influence of operating and electrochemical parameters on PEMFC performance through simulation experiments, which has potential application value in engineering. In order to meet the requirements of high-quality publication of the "membranes" journal, it is recommended to consider the following suggestions.
Reply: We thank the reviewer very much for comments and suggestions to improve our work.
1) There is no quantitative data in Abstract Section.
Reply: We thank the reviewer very much for comments. The abstract is modified.
2) Introduction Section needs to be rewritten. In addition to the operating and electrochemical parameters, is other method feasible for improving PEMFC performance? What are the advantages and disadvantages of each? You need to give your own analysis in this section. The following references may have some value and significance, so you can consider quoting them.
Reply: We thank the reviewer very much for comments. We have more text in introduction section..
[1] Review: Modeling and Simulation of Membrane Electrode Material Structure for Proton Exchange Membrane Fuel Cells [J]. Coatings, 2012, DOI: 10.3390/coatings12081145
Reply: Thanks for suggetsions. This paper helped us a lot to improve our MS. We also cited this papers in our MS (Ref 9). We also added few lines regard in revised MS.
[2] A systematic review of machine learning methods applied to fuel cells in performance evaluation, durability prediction, and application monitoring November [J].International Journal of Hydrogen Energy, 2022, DOI: 10.1016/j.ijhydene.2022.10.261
Reply: Thanks for suggetsions. This paper helped us a lot to improve our MS. We also cited this papers in our MS. We added few lines regard in revised MS.
[3] Evolution of atomic-scale dispersion of FeNx in hierarchically porous 3D air electrode to boost the interfacial electrocatalysis of oxygen reduction in PEMFC [J], NANO ENERGY, 2020, DOI: 10.1016/j.nanoen.2020.105734
Reply: Thanks for suggetsions. This paper helped us a lot to improve our MS. We also cited this papers in our MS. We added few lines regard in revised MS.
3) The innovation of this article is not reflected in the first section and needs to be modified.
Reply: We thank the reviewer very much for comments. The first section is modified.
4) More details need to be demonstrated in Model Development Section.
Reply: We thank the reviewer very much for comments. The data is added in revised MS.
5) The method proposed in this paper needs to be compared with the previous literature, otherwise it cannot reflect innovation.
Reply: We thank the reviewer very much for comments. The data is added in revised MS.
6) The Discussion Section needs a separate section.
Reply: We thank the reviewer very much for suggestion. The discusison section is divided into separate section.
7) There is no quantitative data in the Conclusion Section.
Reply: We thank the reviewer very much for comments. The conclusion part is modified
8) There are few references in the last three years.
Reply: We thank the reviewer very much for comments. We have added more references in MS.
9) There is a lack of references in this journal.
Reply: We thank the reviewer very much for comments. We have added more references in MS.
Reviewer 3 Report (New Reviewer)
In the manuscript "Influence of operating and electrochemical parameters on PEMFC performance: A simulation study", the authors have developed a three-dimensional computational fluid dynamic model for evaluating the effect of key operational parameters as well as of key constituents’ properties in the performance of a PEMFC device. The manuscript presents some interesting results, and it can be published after revising these minor points:
1. The manuscript needs a major revision of the English language. Please check the entire manuscript for the several grammatical, syntax, and orthographic errors. There are several.
2. In my opinion, the term “polarized curves” is not correct. Instead, the term “polarization curves” is more accurate.
3. The Butler-Volmer as well as the Nernst equation should be in the manuscript.
Author Response
In the manuscript "Influence of operating and electrochemical parameters on PEMFC performance: A simulation study", the authors have developed a three-dimensional computational fluid dynamic model for evaluating the effect of key operational parameters as well as of key constituents’ properties in the performance of a PEMFC device. The manuscript presents some interesting results, and it can be published after revising these minor points:
Reply: We thank the reviewer very much for comments and suggestions to improve our work.
- The manuscript needs a major revision of the English language. Please check the entire
manuscript for the several grammatical, syntax, and orthographic errors. There are several.
Reply: Thanks for comments. We revised our manuscript with help of experts.
- In my opinion, the term “polarized curves” is not correct. Instead, the term “polarization curves”
is more accurate.
Reply: Thanks for comments. We have changed as per suggestion.
- The Butler-Volmer as well as the Nernst equation should be in the manuscript.
Reply: Thanks for comments. We have revised this section and added equations in MS.
Round 2
Reviewer 1 Report (New Reviewer)
In short, I am satisfied with all the corrections made. Just please check the overall format and improve the grammar to avoid any mistake.
Reviewer 2 Report (New Reviewer)
The authors have addressed all my concerns.
One thing to remind is that the image DPI needs to be improved.
This manuscript is a resubmission of an earlier submission. The following is a list of the peer review reports and author responses from that submission.
Round 1
Reviewer 1 Report
PEM fuel cells are an important technology to enable sustainable mobility, particularly for applications that require high energy. The authors carried out a numerical study of a PEM fuel cell with the goal to reveal internal processes. Therefore, they calibrated a CFD channel model to an experimental polarization curve without revealing its origin.
Though this is nice work, the following reasons hinder me from accepting this paper:
- The paper reads like a project report that was carried out with ANSYS. I see no learnings or methodological benefits that are of interest for the community. The conclusions that are drawn by the authors are mostly well known since many years.
- The English language is far from being sufficient.
- From my point of view, the paper has not a sufficient quality for a journal publication.
Reviewer 2 Report
1)Please highlight the novelty of this study in your manuscript,
2)Add the conditions of the simulation to all figures,
3)The abstract and the conclusion should be more quantitative,
4)Please add Recommandations by the end of this wor work,
5)References are presented in alphabetic order, wich is different from the form requested by the journal,
6)This maniscript needs to be formatted carefully and the language should be re-approved by a native speaker/Grammarly checked.
Round 2
Reviewer 1 Report
improve obviously unsuitable papers
Author Response
Dear Editor,
Thank you very much for your kind consideration our manuscript submitted to Membranes (Manuscript ID:-2027042). We recieved detail revision for our manuscript and tried to revise manuscript very carefully according to the reviewers’ comments, with all the revisions highlighted in blue, and the following is the point-to-point response to the reviewers’ comments. Finally, my co-authors and I appreciate the time and attentions the reviewers gave to our manuscript to improve the quality of our submission.
Further, the name of Prof. Jong Hwan Lim, Jeju National University South Korea has been replaced with his student Mr. Wajid Ali upon his request. Thank you very much for your kind consideration and we are looking forward to hearing from you soon.
Sincerely yours,
Dr. Khalid Hussain Thebo
IMR, Chinese Academy of Sciences
Beijing 100049, People’s Republic of China
Tel: 008613041129276
E-mail: khalidthebo@yahoo.com / khalid14b@imr.a.c.cn

Reviewer 2 Report
No response to my last comments.
Author Response

(The authors gave the same response as above.)
